# Growth of bilayer MoTe$_2$ single crystals with strong non-linear Hall effect

Teng Ma[1,2,7], Hao Chen[1,3,7], Kunihiro Yananose [4,7], Xin Zhou [1], Lin Wang [1], Runlai Li [1], Ziyu Zhu[1], Zhenyue Wu[1], Qing-Hua Xu [1], Jaejun Yu [4], Cheng Wei Qiu[5], Alessandro Stroppa [6] ✉ & Kian Ping Loh [1,2,3] ✉

The reduced symmetry in strong spin-orbit coupling materials such as transition metal ditellurides (TMDTs) gives rise to non-trivial topology, unique spin texture, and large charge-to-spin conversion efficiencies. Bilayer TMDTs are non-centrosymmetric and have unique topological properties compared to monolayer or trilayer, but a controllable way to prepare bilayer MoTe$_2$ crystal has not been achieved to date. Herein, we achieve the layer-by-layer growth of large-area bilayer and trilayer 1T′ MoTe$_2$ single crystals and centimetre-scale films by a two-stage chemical vapor deposition process. The as-grown bilayer MoTe$_2$ shows out-of-plane ferroelectric polarization, whereas the monolayer and trilayer crystals are non-polar. In addition, we observed large in-plane nonlinear Hall (NLH) effect for the bilayer and trilayer T$_d$ phase MoTe$_2$ under time reversal-symmetry conditions, while these vanish for thicker layers. For a fixed input current, bilayer T$_d$ MoTe$_2$ produces the largest second harmonic output voltage among the thicker crystals tested. Our work therefore highlights the importance of thickness-dependent Berry curvature effects in TMDTs that are underscored by the ability to grow thickness-precise layers.

Transition-metal ditellurides (TMDTs) show rich quantum phases, including Weyl semimetal (T$_d$ phase) and quantum spin hall insulator for the bulk and monolayer, respectively[1–6]. MoTe$_2$ is particularly interesting because it can exist as the semiconducting 2H or the semi-metallic 1T′ phase at room temperature, and at a critical temperature of 260 K, the monoclinic (space group, $P2_1/m$) 1T′ MoTe$_2$ transforms into an orthorhombic polar T$_d$ (space group, $Pmn2_1$) phase, which is a rare example of a polar metal with superconductivity. Few-layer TMDTs have higher density of states, charge mobilities and stability than their monolayer counterparts[1–6]. Although bulk 1T′ MoTe$_2$ is topologically trivial, scaling the material to ultrathin limit lowers the symmetry and

enables a new form of canted spin Hall effect, characterized by concurrent in-plane and out-of-plane spin polarizations[7,8]. Low symmetry MoTe$_2$ possesses non-trivial spin textures and spin-polarized surface states that can be useful in deterministic spin-orbit torque switching.

For both T$_d$ and 1T′ phase, the Berry curvature dipole can be highly sensitive to thickness-dependent symmetry effects that vary markedly between monolayer and bilayer. These ultrathin TMDTs provide a materials platform for studying topological phase transitions, interlayer interactions, correlated electronic phases[1], moiré superlattices[2] and symmetry-dependent spin-orbit physics[3–5]. A scalable approach to fabricate monolayer and bilayer TMDTs is thus very

[1]Department of Chemistry, National University of Singapore, Singapore 117543, Singapore. [2]Department of Applied Physics, Hong Kong Polytechnic University, Hung Hom, Kowloon, Hong Kong, P. R. China. [3]Centre for Advanced 2D Materials, National University of Singapore, 6 Science Drive 2, Singapore 117546, Singapore. [4]Center for Theoretical Physics, Department of Physics and Astronomy, Seoul National University, Seoul 08826, Republic of Korea. [5]Department of Electrical and Computer Engineering, National University of Singapore, Singapore 117583, Singapore. [6]Consiglio Nazionale delle Ricerche, Institute for Superconducting and Innovative Materials and Devices (CNR-SPIN), c/o Department of Physical and Chemical Sciences, University of L'Aquila, Via Vetoio I-67100 Coppito, L'Aquila, Italy. [7]These authors contributed equally:Teng Ma, Hao Chen, Kunihiro Yananose. ✉e-mail: alessandro.stroppa@spin.cnr.it; chmlohkp@nus.edu.sg

useful to investigate how charge, topology and symmetry change giving rise to new quantum phases.

Chemical vapor deposition (CVD) has been extensively used to grow layer-controlled transition metal dichalcogenides (TMDCs) single crystals and films. Different approaches such as reverse-flow chemical vapour epitaxy[9,10] and enhanced nucleation using growth promoters[11,12] have been reported for the growth of MoS$_2$, WS$_2$, MoSe$_2$, and WSe$_2$[13]. However, owing to the small electronegativity difference (0.3 eV) between Mo/W and Te in TMDT, the direct growth of 1T' stoichiometric ditellurides with precise layer control is challenging[14,15]. Although several strategies had been developed, such as phase engineering[16], tellurization of molybdenum oxide[17], molybdenum films[18], ammonium molybdate tetrahydrate ((NH$_4$)$_6$Mo$_7$O$_{24}$·4H$_2$O)[19,20] (abbreviated as AHM), MoS$_2$ or WS$_2$[21], the grown MoTe$_2$ films are invariably non-uniform in terms of thickness and are of low crystalline quality[22]. A scalable synthesis method to prepare single crystalline TMDTs with precise layer control remains extremely challenging.

Herein, we demonstrate the centimeter-sized growth of high-quality bilayer 1T' MoTe$_2$ films and trilayer single crystals using layer-by-layer homoepitaxy. The monolayer MoTe$_2$ obtained at the first stage serves as the growth template to enable layer-by-layer growth. The CVD-grown bilayer 1T' MoTe$_2$ exhibit strong room-temperature out-of-plane ferroelectricity polarization, while the trilayer is non-polar. Both the bilayer and trilayer MoTe$_2$ single crystals show large in-plane nonlinear Hall (NLH) effect. Bilayer T$_d$ MoTe$_2$ exhibits a second harmonic output voltage of 125 μV at an input current of 97 μA and with an in-plane nonlinear Hall magnitude up to $7 \times 10^{-3}$ μm·V$^{-1}$, these are among the highest reported values[23–25] for two-dimensional materials so far.

## Results and Discussion

### Layer-by-layer growth of MoTe$_2$ single crystals and films

MoTe$_2$ single crystals were grown on SiO$_2$/Si substrate by a sequential two-stage CVD process at ambient pressure (Fig. 1a–e and Methods). Prior to the growth, 5% sodium chlorate (SC) solution was spin-coated onto the substrate used to enhance the adsorption of growth precursors[26]. In the first-stage growth, monolayer MoTe$_2$ single crystal were grown using AHM precursor and tellurium powder at 700 °C, and in the atmospheric pressure of pure argon. In the absence of H$_2$ flow, only AHM precursor supplied the Mo precursors needed for MoTe$_2$ growth. This allows the concentration of the active MoTe$_2$ species to be sufficiently low to enable self-limiting monolayer growth (Fig. 1f). To initiate second layer growth, a low flow rate of H$_2$ is introduced into the reaction zone to reduce the MoO$_3$ precursors into volatile MoO$_{3-x}$ species, this increases the Mo supersaturation and initiates the epitaxial growth of the second layer (Fig. 1g, h). Increasing hydrogen flow rate increases the concentration of Mo precursors on the surface and allows the growth kinetics to be changed from attachment-limited to

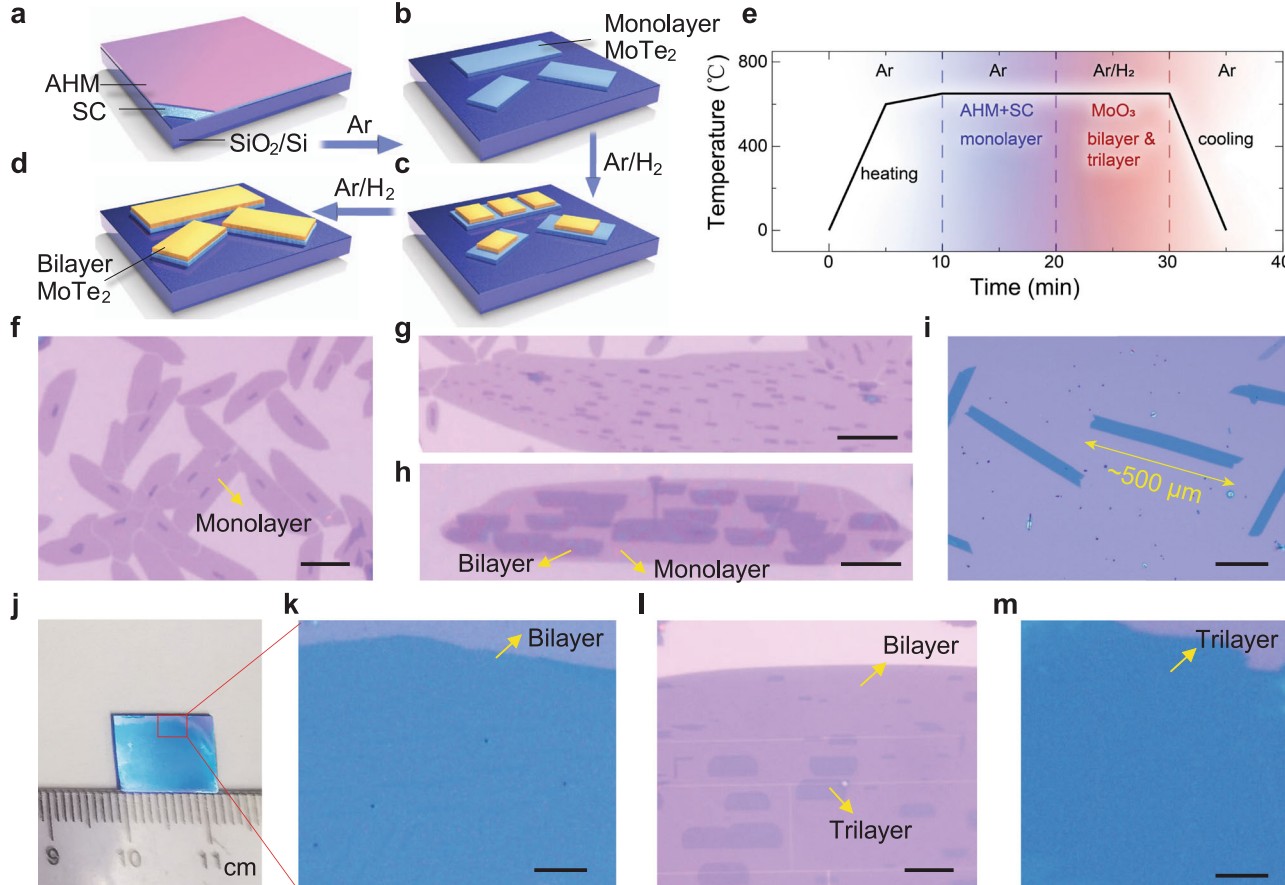

**Fig. 1 | Layer-by-layer growth of centimetre-scale MoTe$_2$ films. a–e,** Schematic (**a-d**) and growth parameters (**e**) for the growth of MoTe$_2$ single crystals. The temperature profile in **e** indicates the temperature at SiO$_2$/Si substrates. The blue and red shaded areas represented the growth stage for monolayer, bilayer, and trilayer growth, respectively. AHM and SC in **a** and **e** represented the Mo precursor (ammonium molybdate tetrahydrate, (NH$_4$)$_6$Mo$_7$O$_{24}$·4H$_2$O) and growth promotor (sodium chlorate), respectively, which were used for the monolayer MoTe$_2$ growth. **f–i** Optical images of MoTe$_2$ single crystals at different growth stages: monolayer dominated (**f**), initial nucleation on the monolayer (**g**), aligned monolayer single crystals on top of the monolayer (**h**), bilayer MoTe$_2$ single crystals with length of ~500 μm (**i**). The scale bars in **f–h** are 10 μm. The scale bar in **i** is 200 μm. **j, k** Centimetre-sized polycrystalline bilayer film (**j**), and enlarged area of the red box in **j** (**k**). The scale bars in **k** is 200 μm. **l, m** Optical images of aligned monolayer MoTe$_2$ domains on top of bilayer film (**l**) and large-area polycrystalline trilayer film (**m**). The scale bars in **l** and **m** are 10 μm and 200 μm, respectively.

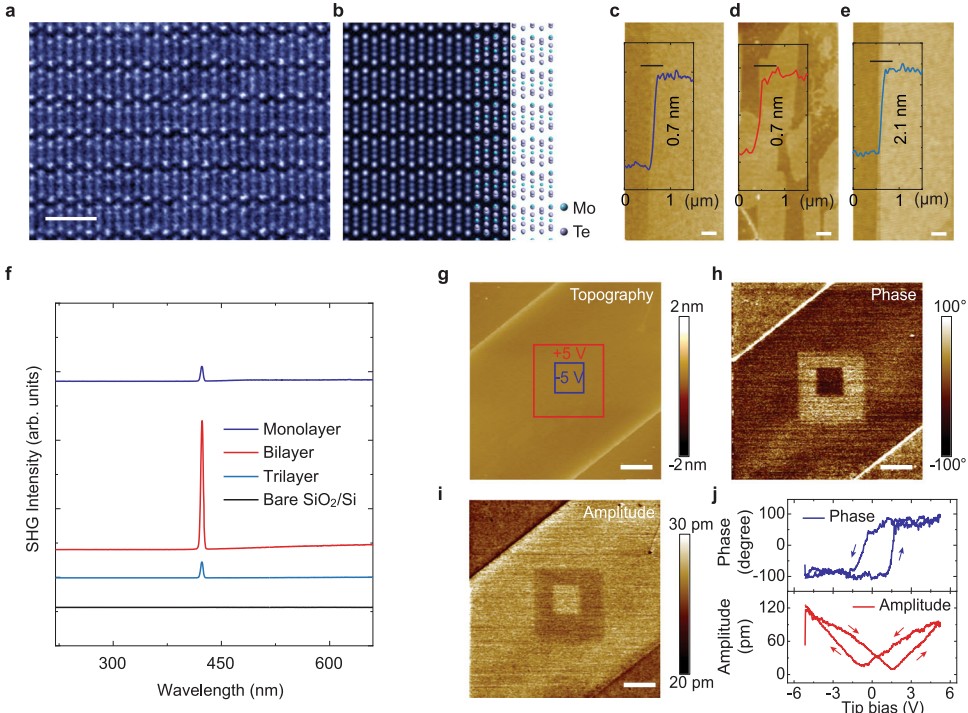

**Fig. 2 | Atomic structure and layer-dependent inversion symmetry breaking of MoTe₂ film. a, b** Aberration-corrected scanning transmission electron microscopy images (**a**) and corresponding simulated atomic structure (**b**) of bilayer 1T′ MoTe₂. The scale bar is 1 nm. **c–e** Atomic force microscope images of MoTe₂ with different layers, showing the thickness of monolayer with 0.7 nm (**c**), epitaxially grown 0.7 nm-thick second layer on top of the monolayer (**d**), and 2.1 nm-thick trilayer (**e**). All the scale bars are 1 μm. **f,** Second-harmonic generation measurement of MoTe₂ with different layers. **g–i** Topography (**g**), phase (**h**), and amplitude (**i**) images of bilayer 1T′ MoTe₂ single crystals recorded after polarization switching with DC bias +5 V in the central region of 5 μm followed by −5 V in the central region of 2 μm. All the scale bars are 2 μm. **j** Phase (blue curve) and amplitude (red curve) signals as a function of the tip voltage for the selected points, showing local hysteresis and butterfly loops.

diffusion-limited growth (Supplementary Fig. 1). Using this method, the size of the bilayer single-crystal MoTe₂ domain can reach up to ~500 μm (Fig. 1i). By extending the growth time, trilayer single-crystal domain with lateral width larger than 100 μm could be grown on the bilayer crystals (Supplementary Fig. 2).

To grow a continuous monolayer MoTe₂ film over large area (Supplementary Fig. 3), a ~200 nm-thick AHM precursor film was spin-coated on the substrate. By extending the growth time, the aligned isolated single crystals merge with each other, and centimetre-sized bilayer polycrystalline films (Fig. 1j, k) and large-scale trilayer films could be fabricated (Fig. 1l, m).

## Structural characterization

To confirm the atomic structure of as-synthesized MoTe₂, aberration-corrected scanning transmission electron microscopy (STEM) were carried out on the as-grown monolayer, bilayer and trilayer films. Details explaining how mono, bi, and trilayers are differentiated are provided in Supplementary Figs. 4–8. Figure 2a shows the atomic high-angle annular dark field (HAADF) image of bilayer 1T′ MoTe₂, which is distinguishable from the monolayer and trilayer. The quasi-1D molybdenum-tellurium zigzag chains are along the a-axis of the unit cell and connected by Te atoms in between, as indicated in the atomic structural model shown in Fig. 2b and Supplementary Fig. 7. STEM imaging of the bilayer and trilayer MoTe₂ reveals that the stacking order is 1T′, which is characterized by a distorted unit cell (Fig. 2a and Supplementary Fig. 4). The STEM-HAADF image shows excellent agreement with the simulated models of the bilayer and trilayer. Powder X-ray diffraction (XRD) collected from the as-synthesized MoTe₂ films also confirms the 1T′ phase. The sharp XRD peaks with a full-width-at half-maximum of ~0.1° indicate the

high crystalline qualities, and the predominance of the (00 l) peaks suggests that the MoTe₂ crystals are textured in (00 l) (Supplementary Fig. 9). The monolayer, bilayer, and trilayer can be distinguished clearly by the thickness of their edges using atomic force microscope (AFM). Figure 2c–e shows the AFM image of MoTe₂ monolayer on SiO₂/Si (Fig. 2c), aligned MoTe₂ crystals on the monolayer (Fig. 2d), and the trilayer film (Fig. 2e). The layer thickness and uniformity of the samples were verified by Raman spectroscopy (Supplementary Fig. 10) and mapping (Supplementary Fig. 11). Especially, the finger-print Raman peak at 269 cm⁻¹ distinguishes monolayer MoTe₂ from thicker layers, the peak position redshifts to 267, and 265 cm⁻¹ for bilayer and trilayer, respectively, which can be explained by the increased dielectric screening of the long-range Coulomb interaction in thicker MoTe₂[27].

Second harmonic generation (SHG) spectroscopy was used to identify the layer-dependent inversion symmetry breaking for 1T′ MoTe₂ crystal (Fig. 2f). The as-grown bilayer crystal exhibits a strong SHG intensity because it belongs to the acentric *Pm* space group, whereas monolayer and trilayer 1T′ MoTe₂ belongs to the *P2₁/m* symmetric monoclinic space group and gives negligible SHG signal. To verify the existence of the stable and switchable polarization in the bilayer 1T′ MoTe₂ at room temperature, piezoresponse force microscopy (PFM) was used to detect the ferroelectric domains at the nanoscale (Fig. 2g–j). To demonstrate the electrical switching of ferroelectric domains, we first scanned the vertical PFM signals of the initial state, where the phase and amplitude signals are basically uniform in the whole area, suggesting a single-domain state (Supplementary Fig. 12). A DC tip bias of +5 V was then used to scan the central regions of 5 μm in the respective film, followed by the application of −5 V tip bias to scan the central area of 2 μm. This produces box-in-box

domains and domain walls that show contrast in the respective phase and amplitude images, confirming the polarization switching of the ferroelectric domains (Fig. 2h, i and Supplementary Fig. 13). PFM-based hysteresis loop measurements shown in Fig. 2j reveal the characteristic butterfly loops of amplitude signal and 180° reversal of phase signal during switching. PFM measurement on a monolayer (Supplementary Fig. 14) and trilayer (Supplementary Fig. 15) 1T′ MoTe$_2$ single crystals did not produce any polarization switching due to their non-polar nature[28].

## Nonlinear Hall effect

The ordinary Hall effect is observed in the presence of a magnetic field normal to the direction of the electric field, while the anomalous Hall effect[29] can manifest in magnetic materials. In the linear regime, both these effects require time-reversal symmetry breaking. In the non-linear regime, however, a second-order Hall voltage can be measured in time-reversal symmetric condition[30–35] in materials with broken inversion symmetry. The second harmonic Hall effect was observed in bilayer WTe$_2$[25], few-layer WTe$_2$[24] and bulk TaIrTe$_4$[36] and the intrinsic origins can be traced to the presence of Berry curvature dipole in these inversion asymmetric materials. Berry curvature dipole $D_{\alpha\beta}$ is a rank-2 pseudotensor[30] determined from the reciprocal space properties. In both T$_d$ MoTe$_2$ and WTe$_2$, the out-of-plane second harmonic Hall voltage is related to the Berry curvature dipole components $D_{xy}$ and $D_{yx}$ which are allowed in both bulk and thin layers[23]. On the other hand, $D_{xz}$ component which gives rise to the in-plane second harmonic effect vanishes in bulk crystal possessing the xz- mirror glide plane or the 2-fold z-screw axis. However, these symmetries, including the z-direction translation, is broken in thin layers, thus $D_{xz}$ component of Berry curvature dipole can be finite in thin layers such as bilayer and trilayer T$_d$ phase, which is the subject of our investigation.

We have studied the Berry curvature dipole and its dependence on layer thickness using density functional theory (DFT) calculation. Figure 3a, b shows the calculated band structures of the MoTe$_2$ T$_d$ bilayer and trilayer, respectively. Energy levels are referred to the Fermi level of undoped system, $E_{F,0}$. Based on the carrier densities measured experimentally, we estimated the Fermi levels as 0.28 eV and 0.24 eV for the bilayer and trilayer, respectively (Supplementary Fig. 16). The Berry curvature distributions in k-space at these energy levels are plotted in Fig. 3c, e for bilayer and trilayer T$_d$ MoTe$_2$, respectively. The opposite signs across yz-mirror plane are consistent with the finite $D_{xz}$ component. The change in Berry curvature dipoles with Fermi level are shown in Fig. 3g, h, from which we can extract $D_{xz}$ as 0.040 nm and 0.155 nm for bilayer and trilayer T$_d$ MoTe$_2$, respectively.

The absence of inversion symmetry splits the spin-degeneracy of bands by spin-orbit coupling. Furthermore, the interlayer interaction modifies the bands of each layer. As a result, complex crossing and anti-crossing points that contribute to Berry curvature appear in the band structure[37]. For the bilayer, four crossing points between the lowest and the second-lowest conduction bands are found at ($\pm 0.2034\frac{2\pi}{a}$, $\pm 0.0143\frac{2\pi}{b}$) in the Brillouin zone (inset of Fig. 3a). The Berry curvature near the crossing points calculated by the fixed-number-of-occupied-bands scheme is shown in Fig. 3d. For the trilayer, anti-crossing points between the second- and the third-lowest conduction bands are found at ($\pm 0.1149\frac{2\pi}{a}$,0) as highlighted in the inset of Fig. 3b. The Berry curvature near the anti-crossing point of trilayer is shown in Fig. 3f. We note that, the points where the Berry curvature has a large magnitude, so-called Berry curvature "hot spots" coincide with the crossing/anti-crossing points. It can be seen that the energy level of the crossing points of the bilayer (0.127 eV with respect to $E_{F,0}$) agrees with the peak position of the $D_{xz}$ shown in Fig. 3g. For the trilayer, the anticrossing points give

rise to peaks near 0.12 eV in $D_{xz}$ (Fig. 3h). This suggests that the origin of the peaks in the Berry curvature dipole is related to the band crossings and anti-crossings.

To measure the nonlinear Hall effect in bilayer and trilayer MoTe$_2$, the CVD-grown MoTe$_2$ crystal is patterned with Hall bar contacts (see inset of Fig. 4a) using standard electron beam lithography and then encapsulated with a thin hexagonal boron nitride (hBN) flake. The entire fabrication process was carried out under an inert atmosphere to prevent sample degradation. The only symmetry in the few-layer T$_d$ MoTe$_2$ is the mirror plane perpendicular to the a-axis. In this case, the nonlinear Hall response signal is largest when the current is along the a-axis (see Methods for details). As shown in Supplementary Fig. 7, the MoTe$_2$ lattice is oriented such that the a-direction points along the long-axis of the rectangular flake. Therefore, we can make use of this anisotropic growth to pattern the source drain/Hall contacts along the a or b-axis of the MoTe$_2$ crystal, respectively. Electrical measurements were carried out in a temperature-variable helium cryostat in high vacuum. The upswing in resistance at low temperature for few-layer samples has been reported previously for ultrathin MoTe$_2$[38–40]. In addition, we find that the substrate also plays a crucial role in modifying the electronic band structure of MoTe$_2$[41–43]. A detailed discussion can be found in Supplementary Information.

The magnetoresistivity $\rho_\parallel$ (red line) and Hall resistivity $\rho_\perp$ (blue line) curves of bilayer MoTe$_2$ are shown in Fig. 4b at a base temperature of 1.6 K. We can extract a carrier (electron) density of $8.23 \times 10^{13}$ cm$^{-2}$ at base temperature (Supplementary Fig. 16), which indicates that the sample is in a heavily doped state. Hall measurements are performed at the fundamental frequency with current $I$ applied along a-axis and four terminal voltages are recorded. The longitudinal voltage $V_\parallel$ increases linearly with current $I$ while first harmonic transverse voltage $V_\perp$ remains small (~1% of $V_\parallel$). The observed finite value of $V_\perp$ could arise from misalignment of the electrodes with the crystal axes (typically of the order of ~1°) and the intrinsic resistance anisotropy of the material (Supplementary Fig. 17 and Table 1).

Next, we examine the second harmonic transverse voltage $V_\perp^{2\omega}$ as function of applied longitudinal voltage $V_\parallel^\omega$. An AC current is applied from the source (S) to the drain (D) electrode (input current) and the differential voltage is measured between the A and B electrodes (output voltage). A typical result measured at 40 K is shown in Fig. 4c and results for higher temperatures (larger than 100 K) are shown in Supplementary Fig. S18. The linear dependence of $V_\perp^{2\omega}$ on the square of $V_\parallel^\omega$ demonstrates the second harmonic origin of detected signal. In addition, we employ two opposite measurement setups where red and blue dots represent data acquired from forward and backward current, respectively. The second-order Hall signal switches sign when the current orientation and the Hall probes connections are reversed simultaneously. This sign reversal behavior further consolidates its second order origin because linear Hall effect would vanish under Onsager reciprocal relation while second-order one does not need to vanish[30,44]. Additional data of the observed second-order signal, which can be found in Supplementary Figs. 19–21, further clarifies the origin and excludes the thermal effect.

Based on the Hall bar geometry, we can obtain electric fields for each quantity: $E_\perp^{2\omega} = \frac{V_\perp^{2\omega}}{L_\perp}$ and $E_\parallel = \frac{V_\parallel}{L_\parallel}$ where $L_\parallel \approx 2.4$ μm and $L_\perp \approx 1.7$ μm are longitudinal and transverse lengths of the bilayer Hall bar device, respectively. At fixed temperatures ranging from 10 K to 100 K, $E_\perp^{2\omega}$ depends linearly on $E_\parallel^2$ as shown in Fig. 4d. Similar linear dependence of $E_\perp^{2\omega}$ on $E_\parallel^2$ is also obtained for trilayer MoTe$_2$ sample. A measure of the magnitude of the nonlinear Hall effect (NLHE) can be extracted from the slope of the linear plot, $\frac{E_\perp^{2\omega}}{E_\parallel^2}$. We can rewrite $\frac{E_\perp^{2\omega}}{E_\parallel^2} = \frac{\sigma_{NLH}}{r\sigma E_\parallel}$, where $\sigma_{NLH}$, $\sigma$, and $r(\equiv \frac{\rho_a}{\rho_b})$ denote nonlinear Hall conductivity, conductivity

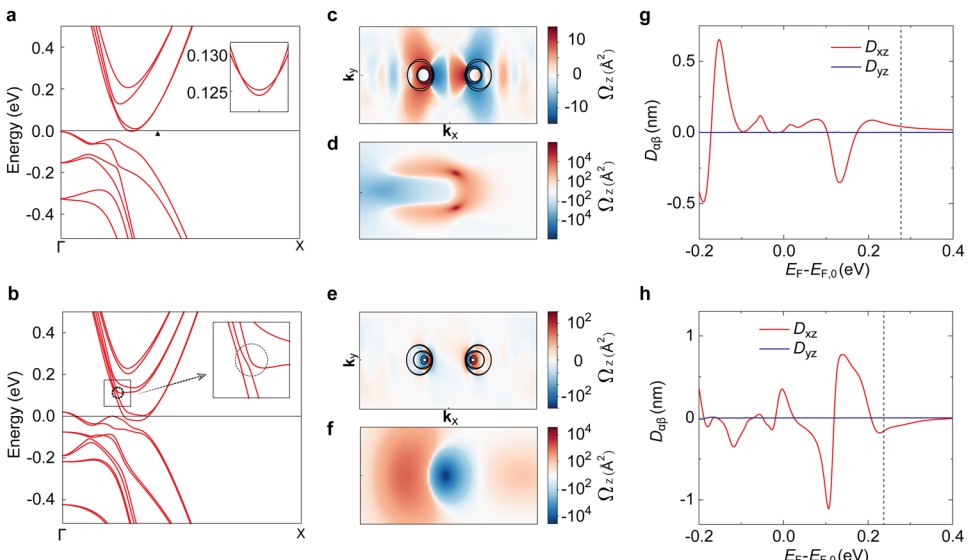

**Fig. 3 | Calculated electronic band structure and Berry curvature of bilayer and trilayer MoTe₂. a**, **b** Band structure of the MoTe₂ T$_d$ bilayer (**a**) and trilayer (**b**), respectively. The small triangle in **a** indicates $k_x = 0.2034 \frac{2\pi}{a}$ through which the band path of inset passes. The inset in **a** show the crossing points in the bands along the path connecting two points $(0.2034 \frac{2\pi}{a}, \pm 0.0228 \frac{2\pi}{b})$ which is perpendicular to ΓX line. The small circle in **b** indicates the anti-crossing point that is magnified in **b**. **c**–**f** (**c**) and (**e**) shows Berry curvature (z-component) distribution in momentum space at specific energy levels (0.28 and 0.24 eV) for MoTe₂ bilayer and trilayer,

respectively. The black solid lines are Fermi lines. **d** and **f** are the zoom-in of Berry curvature near corresponding crossing/anti-crossing points in the areas defined by intervals $k_x \in [0.16 \frac{2\pi}{a}, 0.24 \frac{2\pi}{a}]$ and $k_y \in [-0.04 \frac{2\pi}{b}, 0.04 \frac{2\pi}{b}]$ for **d** and $k_x \in [0.075 \frac{2\pi}{a}, 0.155 \frac{2\pi}{a}]$ and $k_y \in [-0.04 \frac{2\pi}{b}, 0.04 \frac{2\pi}{b}]$ for **f**. **g**, **h** Berry curvature dipole as function of chemical potential for bilayer (**g**) and trilayer (**h**), respectively. Dotted lines in **g** and **h** denotes Fermi energy of 0.28 eV and 0.24 eV for bilayer and trilayer samples, respectively.

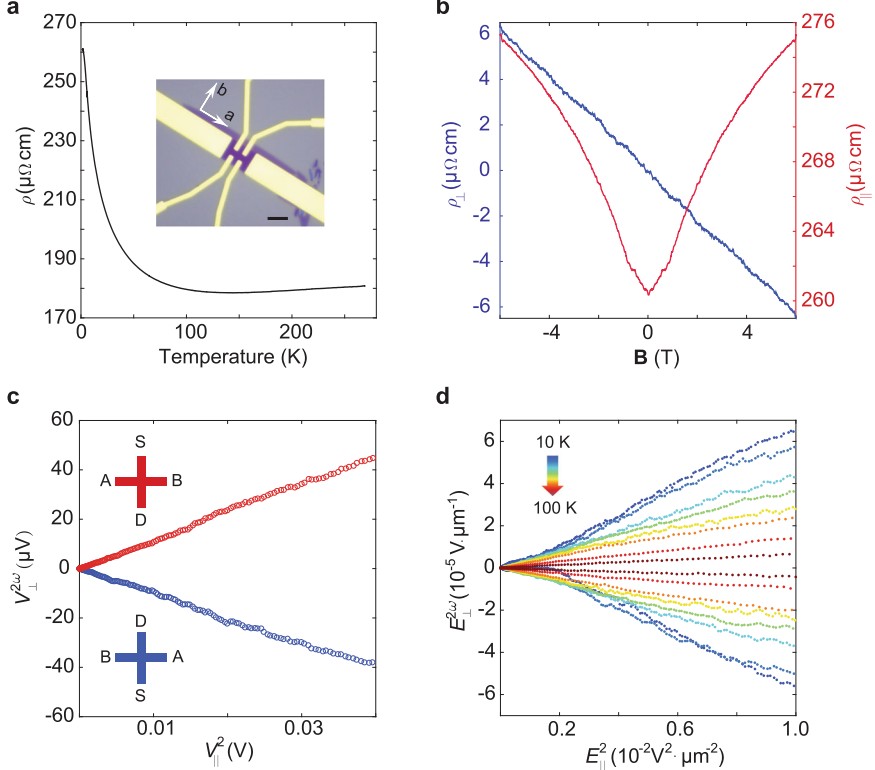

**Fig. 4 | Second harmonic Hall effect in bilayer MoTe₂. a** Temperature dependence of resistivity of bilayer MoTe₂. Inset: optical images of Hall bar device of as-grown bilayer MoTe₂, a and b denote the crystal lattice orientation. The scale bar is 5 μm. **b** Longitudinal resistivity $\rho_\parallel$ and Hall resistivity $\rho_\perp$ measurement of sample at 1.6 K. **c** Second-harmonic Hall voltage $V_\perp^{2\omega}$ as function of square of longitudinal

voltage $V_\parallel^2$ measured at 40 K. Red and blue cross shows opposite measurement setup. (S, D, A, and B denotes source, drain, Hall probe A and B respectively). **d** Linear dependence of $E_\perp^{2\omega}$ on $E_\parallel^2$ at temperatures ranging from 10 to 100 K (in order of 10, 20, 30, 40, 50, 60, 80, and 100 K as indicated by arrows).

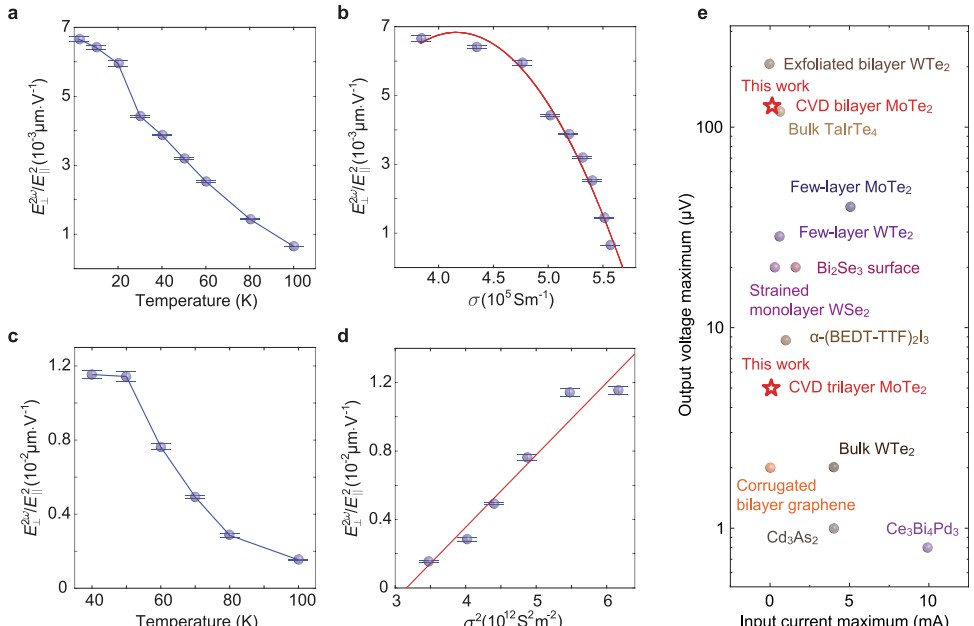

**Fig. 5 | Temperature dependence of the 2nd harmonic Hall effect. a, b** $E_\perp^{2\omega}$ of bilayer MoTe$_2$ as a function of temperatures (**a**) and conductivity (**b**). **c, d** $E_\perp^{2\omega}$ of trilayer MoTe$_2$ as function of temperature (**c**) and square of conductivity (**d**). Blue lines in **a** and **c** are guides to the eye. Red curves in **b** and **d** are parabolic and linear fits to experimental data for a bilayer and trilayer sample, respectively. Symbols for experimental data are shown with error bars (some are smaller than symbols) obtained by linear regression for the relation of $E_\perp^{2\omega}$ on $E_\parallel^2$ at each temperature. **e**, Input current-output nonlinear Hall effect voltage map of materials with non-linear Hall effect for comparison.

(along a-axis) and resistivity anisotropic ratio, respectively. Here $\sigma_{NLH}$ is derived from second-order susceptivity tensor (see Methods). Since $J = \sigma E_\parallel$, nonlinear Hall conductivity normalized by current density ($\frac{\sigma_{NLH}}{J}$) can be used to describe the magnitude of the NLH effect, thus it follows that $\frac{E_\perp^{2\omega}}{E_\parallel^2}$ can also be used as a parameter to express the magnitude of NLHE. $\frac{E_\perp^{2\omega}}{E_\parallel^2}$ of bilayer and trilayer at different temperatures are shown in Fig. 5a, c, respectively. From these, we can investigate the scaling relationship between $\frac{E_\perp^{2\omega}}{E_\parallel^2}$ and conductivity for the bilayer and trilayer samples, as plotted in Fig. 5b, d, respectively.

The relation between NLH magnitude and conductivity has been formulated by Du et al.[45] as:

$$\frac{V_y^N}{\left(V_x^L\right)^2} \simeq C_1 \sigma_{xx0}^{-1} \sigma_{xx}^2 + (C_2 + C_4 - C_3)\sigma_{xx0}^{-2}\sigma_{xx}^2$$
$$+ (C_3 - 2C_4)\sigma_{xx0}^{-1}\sigma_{xx} + C_4. \quad (1)$$

where the left term $\frac{V_y^N}{\left(V_x^L\right)^2}$ is proportional to the $\frac{E_\perp^{2\omega}}{E_\parallel^2}$ only by a geometric factor. Here $\sigma_{xx}$ is the longitudinal conductivity while $\sigma_{xx0}$ is the conductivity at zero temperature limit. Parameters $C_{1,2,3,4}$ include contribution from intrinsic ($C^{in}$), side-jump ($C^{sj}$), intrinsic skew-scattering ($C_i^{sk,1}$), and extrinsic skew-scattering ($C_{ij}^{sk,2}$) as follows:

$$C_1 = C^{sk,2}, C_2 = C^{in} + C_0^{sj} + C_{00}^{sk,1}, C_3 = 2C^{in} + C_0^{sj} + C_1^{sj} + C_{01}^{sk,1}, C_4 = C^{in} + C_1^{sj} + C_{11}^{sk,1}$$

The NLH magnitudes are strong for our bilayer and trilayer samples at temperature range below 100 K. At this temperature range, the conductivity of our bilayer sample is ~$10^5$ S m$^{-1}$.

In the literature, the reported conductivity of 2D materials with NLHE ranges from ~$10^5$ S m$^{-1}$ (for few-layer WTe$_2$[24]) to ~$10^8$ S m$^{-1}$ (for thick MoTe$_2$ and WTe$_2$[23]). Thus, the conductivity of our bilayer sample is several orders of magnitude smaller than previously reported 2D

materials with NLHE and it is an order of magnitude smaller than that of our trilayer sample (~$10^6$ S m$^{-1}$).

At such low conductivity, both the first and second order term of $\sigma_{xx}$ as described in Eq. (1) contributes to NLH magnitude. The NLH magnitude versus conductivity plot for the bilayer sample can be fitted with a parabolic curve described by Eq. (1). The detailed fitting parameters are presented in Supplementary Information. To quantitatively obtain each parameters $C_{1,2,3,4}$ requires additional data from samples with various thicknesses[41], which is beyond the scope of the current work.

In contrast to bilayer (Fig. 4a), the trilayer sample shows metallic behavior as resistivity decreases with decreasing temperature (Supplementary Fig. 22). A typical second harmonic Hall response of trilayer is displayed in Supplementary Fig. 23. In Fig. 5d, the NLHE magnitude of trilayer sample shows a linear dependence on the square of conductivity which can be described as Eq. (2) below.

$$\frac{E_\perp^{2\omega}}{E_\parallel^2} = \alpha\sigma^2 + \beta \quad (2)$$

where α and β are constants. Such a relationship agrees well with previously reported NLHE materials as second order σ term dominates in highly conducting samples[24]. Different from bilayer, the observed scaling relation is simpler for trilayer, where NLHE magnitude only contains contributions from intrinsic ($\sigma^0$) and skew scattering (scales as $\sigma^2$).

For the intrinsic part, if we neglect side jump contribution, an estimate of Berry curvature dipole could be obtained from the intercept $|\beta| \sim 1.34 \times 10^{-2}$ μm·V$^{-1}$ in Fig. 5d. Detailed fitting process and parameters can be found in Supplementary Information. The extracted experimental dipole is $D \sim \beta \cdot \frac{\epsilon_F}{e} \cdot \frac{r}{\pi} \sim 0.3$ nm, which is in the same order as the theoretically calculated Berry curvature dipole of 0.155 nm at the chemical potential at $\epsilon_F \sim 0.24$ eV, as shown in Fig. 3h. It is also comparable to values reported for bilayer WTe$_2$[25] (in the range of ~nm), and is an order of magnitude larger than that of few-layer WTe$_2$[24] (~0.1-0.7 Å). It is worth noting that for trilayer sample, NLH magnitude

arising from intrinsic part is almost at the same scale as the conductivity-induced extrinsic part within the measured range. It follows that the intrinsic Berry curvature dipole contribution dominates at low conductivity region ($\sigma < 2 \times 10^6$ S m$^{-1}$) while skew scattering contribution will gradually increase in weightage with increasing conductivity.

Although the NLH behavior of bilayer and trilayer $T_d$ MoTe$_2$ reside in a different regime, the observed in-plane NLH output voltages of both bilayer and trilayer $T_d$ MoTe$_2$ versus the input current are among the highest reported values for 2D materials[23,24,36,46–49], with the exception of exfoliated bilayer WTe$_2$[25], as well as strained[50] and corrugated[51] materials (see Fig. 5e for comparison of current-NLH voltage map and Supplementary Table 2). Bilayer $T_d$ MoTe$_2$ has a maximum output voltage of 125 µV at an input current of 97 µA, with an in-plane nonlinear Hall magnitude up to $7 \times 10^{-3}$ µm·V$^{-1}$. Trilayer $T_d$ MoTe$_2$ has a smaller output voltage of 6.5 µV at an input current of 939 µA compared to bilayer, but due to its higher conductivity, it has a higher in-plane nonlinear Hall magnitude up to $1.2 \times 10^{-2}$ µm·V$^{-1}$. Importantly, the intrinsic NLH due to in-plane Berry curvature dipole only manifests in atomically thin layers, it decreases drastically for thicker layers, i.e., the extracted Berry curvature dipole decreases to 0.047 nm for 8-layer sample, whereby an input current of 810 µA produces only 5.6 µV NLH output (Supplementary Fig. 24).

In conclusion, we have developed a layer-by-layer growth strategy for the synthesis of bilayer and trilayer 1T' MoTe$_2$ single crystals and films. The ability to grow bilayer and trilayer MoTe$_2$ crystals precisely allows us to investigate the symmetry-dependent ferroelectric and non-linear Hall effects that varies dramatically between odd and even layers. Thin-layer $T_d$ MoTe$_2$ phase exhibits a large in-plane nonlinear Hall effect with major contributions from Berry curvature dipole, which is distinct from that of thicker layers dominated by extrinsic charge scattering mechanism. In particular, bilayer $T_d$ MoTe$_2$ exhibits a large non-linear Hall voltage-to-input current ratio compared to thicker layers, which is practically useful for non-linear electrical devices. We believe that applying our two-step growth strategy on vicinal sapphire will enable wafer-scale single-crystal growth, which will be the subject of future studies. This work paves the way for the application of CVD grown, large area, atomically thin $T_d$ MoTe$_2$ phase in nonlinear electrical transport devices.

## Methods

### Layer-by-layer growth of MoTe$_2$ single crystals and films
Before CVD growth, a 5% sodium cholate solution is spin-coated at 2000 rpm for 60 s onto the SiO$_2$/Si substrate, which is followed by heating at 80 °C for 10 min to remove the solvent. After that, saturated solution of AHM in deionized (DI) water is spin-coated at 1000 rpm for 60 s onto the substrate, providing the Mo feedstock at the first stage. The spin-coated substrate is then placed in the centre of a 2-inch CVD tube furnace. MoO$_3$ powder (10 mg) and solid tellurium powder (50 mg) were placed 5 cm and 15 cm upstream from the substrate, respectively. As a result, the temperature at the two locations are 670 and 550 °C, respectively. CVD growth was carried out after the temperature reaches 700 °C in argon atmosphere adjusted to a flow rate of 100 sccm. After 10 min, H$_2$ was bled into the system at a flow rate of 10 sccm to initiate the second layer growth, and the temperature of the SiO$_2$/Si substrate was maintained at 700 °C for 10 min. The temperature of the region at MoO$_3$ and tellurium powder are also maintained at 670 and 550 °C, respectively. To enable the trilayer growth, the H$_2$ flow rate should increase to 15 sccm after finishing the bilayer growth for another 10 min. Finally, the chamber was rapidly cooled to room temperature under a pure Ar flow rate of 100 sccm. The whole CVD reaction was operated at ambient pressure.

### Structural characterization
The morphology, layer number, and quality analysis of the MoTe$_2$ crystals on SiO$_2$/Si substrates were evaluated by optical microscopy (Olympus BX51), Raman spectroscopy (Renishaw, laser wavelength of 532 nm), AFM (Bruker Dimension Icon), and Powder XRD (Bruker D8 Focus Powder X-ray diffractometer using Cu Kα radiation at room temperature). The SHG measurements were performed by a home-built optical setup with a microscope (Nikon, Eclipse Ti) and laser (Coherent, Chameleon Ultra II, laser wavelength of 850 nm). The average laser power was 0.5 mW at the samples. PFM samples were prepared by direct growth of MoTe$_2$ on highly doped Si substrates. The out-of-plane PFM signals were recorded by using a drive frequency of 270 kHz and a drive amplitude of 2000 mV. STEM samples were prepared with a polycarbonate (PC)-assisted dry transfer method to avoid any exposure to water and oxygen. Atomic structures of MoTe$_2$ crystals were characterized using a STEM (JEOL ARM-200F) equipped with an aberration corrector.

### Device fabrication and electrical measurement
CVD-grown 1T' MoTe$_2$ bilayers and trilayers grown on SiO$_2$/Si substrates were identified under optical microscope in glovebox with lower than 0.5 ppm O$_2$ level. Layer number of the sample can be distinguished from optical contrast. Standard electron beam lithography and e-beam evaporation process were performed to introduce 5 nm Ti/70 nm Au contacts. The device was covered with h-BN flake on top for protection. Devices were loaded into a high vacuum helium cryostat and measured using four terminal measurement setup with lock-in amplifiers (Stanford Research Systems Model SR830). Three frequencies (13.373 Hz, 77.77 Hz, and 133.33 Hz) were tested, yielding consistent data (Supplementary Fig. 17). The phase of measured first and second harmonic signals are ~0 and 90 degrees, respectively.

### Nonlinear Hall susceptibility
The second order nonlinear Hall current density $J^{(2)}$ in response to an electric field $E$ can be described by second-order nonlinear susceptivity $\chi^{(2)}$ as $J^{(2)} = \chi^{(2)}EE$ (refs. 52,53). As few layer $T_d$ MoTe$_2$ is in $Pm$ point group symmetry, the susceptivity can be written as:

$$\chi^{(2)} = \begin{pmatrix} d_{11} & d_{12} & d_{13} & 0 & d_{15} & 0 \\ 0 & 0 & 0 & d_{24} & 0 & d_{26} \\ d_{31} & d_{32} & d_{33} & 0 & d_{35} & 0 \end{pmatrix} \quad (3)$$

Here the x, y, and z are chosen to be along the mirror line (b-axis), perpendicular to the mirror line (a-axis) and perpendicular to the plane (c-axis), respectively. When applying in-plane electric field $E = (E_x, E_y, 0)$, the nonlinear Hall current density $J^{(2)}$ is

$$J^{(2)} = \begin{pmatrix} d_{11}E_x^2 + d_{12}E_y^2 \\ 2d_{26}E_xE_y \\ d_{31}E_x^2 + d_{32}E_y^2 \end{pmatrix} \quad (4)$$

Or alternatively, we can obtain the second-order nonlinear electric field from $J^{(2)}$ by Ohm's law $E^{(2)} = \rho J^{(2)}$ and $\rho = \begin{pmatrix} \rho_b & 0 & 0 \\ 0 & \rho_a & 0 \\ 0 & 0 & \rho_c \end{pmatrix}$ is resistivity matrix for anisotropic MoTe$_2$. Thus $E^{(2)} = \begin{pmatrix} \rho_b(d_{11}E_x^2 + d_{12}E_y^2) \\ 2\rho_a d_{26}E_xE_y \end{pmatrix}$. For an in-plane current $J = j\begin{pmatrix} \cos\theta \\ \sin\theta \end{pmatrix}$ of magnitude j and angle $\theta$. The first order electric field is $E = j\begin{pmatrix} \rho_b\cos\theta \\ \rho_a\sin\theta \end{pmatrix}$. The component parallel to $J$ is $E_{\parallel} = j(\rho_b\cos^2\theta + \rho_a\sin^2\theta)$. While the transverse component is

$E_{\perp}^{(2)} = j^2 \rho_b^3 \sin^2\theta(d_{12}r^2\sin^2\theta + (d_{11} - 2d_{26}r^2)\cos^2\theta)$. $r$ is resistivity ratio defined $r \equiv \frac{\rho_a}{\rho_b}$. Thus, we can obtain

$$\frac{E_{\perp}^{(2)}}{E_{\parallel}^2} = \frac{\rho_b\sin\theta\left[d_{12}r^2\sin^2\theta + (d_{11} - 2d_{26}r^2)\cos^2\theta\right]}{\left(\cos^2\theta + r\sin^2\theta\right)^2} \quad (5)$$

This equation describes the angular dependence of the 2$^{nd}$ harmonic Hall effect. At $\theta = 90°$, the largest value is achieved, where $\frac{E_{\perp}^{(2)}}{E_{\parallel}^2} = \rho_b d_{12} = \frac{d_{12}}{r\sigma}$. Since $r$ is dimensionless, $d_{12}$ should have dimension of conductivity over the electric field. Hence, we can define the nonlinear anomalous Hall (NAH) conductivity as $d_{12} = \frac{\sigma_{NAH}}{E_{\parallel}}$.

## First-principles calculations

Density functional theory (DFT) calculations are performed by using Vienna ab initio simulation package (VASP)[54]. Projector augmented wave (PAW) pseudo-potentials[55] were adopted with the inclusion of the spin-orbit coupling. For the exchange-correlation energy, the screened hybrid functional method by Heyd-Scuseria-Ernzerhof (HSE)[56,57] was used in addition to the generalized gradient approximation by Perdew–Burke–Ernzerhof (GGA-PBE)[58]. The range-separation parameter in the HSE functional was fixed at $\mu = 0.114$ Å$^{-1}$. The reduced $\mu$ value in comparison to the standard HSE06 (0.2 Å$^{-1}$) is consistent with the reduced screening in 2 dimensions[59]. Plane-wave basis energy cut-off was set to 400 eV. Regular k-space grid of $\Gamma$-centered $16 \times 8 \times 1$ grid was used. The layered structures were constructed directly from the bulk structures from experiments[60] with the addition of the vacuum layer of about 20 Å. The lattice constants are a = 3.4707 Å and b = 6.3288 Å for $T_d$ phase.

The maximally localized Wannier functions[61] were constructed from the Kohn-Sham wavefunctions and used for the Berry curvature dipole calculations by using WANNIER90 code[62,63]. As the initial projectors, d-orbitals for Mo and p-orbitals for Te were adopted. The Berry curvature dipole $D_{\alpha\beta}$ is evaluated by the following formula.

$$D_{\alpha\beta} = \int d^2\mathbf{k} \sum_n \frac{\partial E_n(\mathbf{k})}{\partial k_\alpha} \Omega_n^\beta(\mathbf{k}) \left( -\frac{\partial f_0(E, E_F, T)}{\partial E} \right)_{E=E_n} \quad (6)$$

where $\alpha$ and $\beta$ represent x, y, or z. $E_n(\mathbf{k})$ is the energy eigenvalue of the band n at $\mathbf{k}$. $f_0$ is the Fermi-Dirac distribution, where the temperature T = 0.004 eV/$k_B$ ≈ 46 K is chosen as a broadening factor. $\Omega_n(\mathbf{k}) = -\text{Im}\langle \nabla_\mathbf{k} u_{n\mathbf{k}} | \times | \nabla_\mathbf{k} u_{n\mathbf{k}}\rangle$ is the Berry curvature. The integral over 2-dimensional Brillouin zone was taken on the $1001 \times 401$ fine grid. For the Berry curvature distributions, WANNIER90 and WANNIERTOOLS[64] were used for the fixed-Fermi-level scheme and fixed-number-of-occupied-bands scheme, respectively.

## Data availability

Relevant data supporting the key findings of this study are available within the article and the Supplementary Information file. All raw data generated during the current study are available from the corresponding authors upon request.

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

## Acknowledgements

K.P.L. and C.-W.Q. acknowledges the support from the National Research Foundation, Prime Minister's Office, Singapore under Competitive Research Program Award NRF-CRP22-2019-0006. K.P.L. acknowledges the support from The Hong Kong Polytechnic University under the PGMS Project Account P0043087 (1.11.56.BDA6). T.M. acknowledges funding support from the Start-up Fund for RAPs under the Strategic Hiring Scheme (Grant No. P0042991, 1-BD5S), The Hong Kong Polytechnic University. A.S. acknowledges support by EC Project HPC-EUROPA3 (H2020-INFRAIA-2016-1-730897). In particular, K.Y. and A.S. acknowledge the computer resources and technical support provided by CINECA HPC centre in Bologna (Italy).

## Author contributions

K.P.L. conceived the project, and T.M. performed the CVD growth, optical, AFM and Raman characterizations. H.C. performed device fabrication and electrical measurements. X.Z. performed the STEM measurements and simulations. K.Y. performed DFT calculations under the supervision of A.S and J.Y. T.M., and L.W. performed the PFM measurements. L.W., Z.W., and C.-W.Q. helped with the analysis of the PFM data. Z.W. and T.M. performed the XRD measurements. Z.Z. and T.M. performed the SHG measurements under the supervision of Q.H.X. R.L. drew the schematics and helped with figure processing. T.M., H.C., and K.P.L. wrote the manuscript with input from all authors.

## Competing interests

The authors declare no competing interests.
