## [Peer Review File · Nature Communications]

Growth of Bilayer MoTe₂ Single Crystals with Strong Non-Linear Hall EffectREVIEWER COMMENTS

Reviewer #1 (Remarks to the Author):

In this work, the authors demonstrate CVD synthesis of MoTe₂ with large domain size (up to 500 μm) and precise control of layer numbers at the ultra-thin limit through a layer-by-layer growth. The synthesized crystals are identified using TEM and Raman spectroscopy. Although the large-area MoTe₂ growth has been achieved using CVD method in literature (Adv. Mater. 28, 9526–9531 (2016), Adv. Mater. 29, 1700704 (2017) and ACS Nano 15, 410–418 (2021)), the precise control of bilayer growth was not presented. The achievement presented in the work provides a platform to explore more interesting properties and applications of the material. Particularly, the nonlinear Hall (NLH) effect observed on bilayer and trilayer Td phase MoTe₂ is interesting and shows the importance of precise thickness control of the growth. More in-depth understanding of some data is required. Below are some questions and comments to be addressed:

1. In the paper, the authors described the synthesized flakes as single crystals with domain size of up to 500 μm (line 100). The optical microscope images show smooth and uniform surface of the flakes. However, the resolution of optical microscope is not high enough to identify the grain boundaries (if they exist) in the sample. More structural evidence (e.g., SAED on different location of the same flake) is required to draw such conclusions.

2. From the optical image, the monolayer growth is preferred along certain directions. What is the reason? What is the stacking order for bilayer and trilayer? Can low frequency Raman signals be observed from the bilayer and trilayer?

3. For the hall bar devices, the authors claimed that the electrodes are patterned along the a- or b-axis of the crystal (line 197). How is this identified?

4. Non-linear Hall effect (NLHE) is observed on MoTe₂ flakes in this work, and opposite NLHE magnitude versus conductivity relationships have been observed for bilayer and trilayer MoTe₂ devices. This is attributed to the low electrical conductivity of bilayer MoTe₂ (line 253). However, the Hall measurements are carried out at low temperature region (<100 K), which coincides with the temperature range of “gap opening” in bilayer MoTe₂ (line 199-204, Fig. 4a). In this scenario, it is difficult to attribute the unique NLHE magnitude vs. conductivity relationship solely to the low magnitude of electrical conductivity.

More control experiments are recommended to identify the origin of this relationship. What if the Hall measurement of bilayer MoTe₂ is carried out at higher temperature where the “gap opening” does not occur (i.e., >100 K)? Will the NLHE magnitude show the same trend as current experiment? Considering the electrical resistivity maintains a low value and does not change much from 100 K to 250 K, the higher temperature region could help identify the role of “gap opening” in this process.

Reviewer #2 (Remarks to the Author):

The authors have studied the nonlinear hall effect NLH in the bilayer and trilayer MoTe₂. By performing transport, PFM measurements they have found interesting thickness dependence ferroelectric domain and second harmonic voltage. They also performed theoretical calculation to claim the major contribution for the large NLH is Berry curvature dipole. However, the experimental results are contradictory to other more convincing works, which is not listed in the reference in this manuscript and this work is not innovative enough, so I do not recommend publishing in Nature Communications. This statement “In Fig. 4a, the increasing resistivity with decreasing temperature suggests a “gap opening” of bilayer sample at low temperature. Previous reports attributed the insulating behavior in few-layer MoTe₂ to either spin-orbit coupling³⁷ or Anderson localization³⁸⁻⁴⁰”

Ref 37 Keum, D. H. et al. Bandgap opening in few-layered monoclinic MoTe₂. Nat. Phys. 11, 473 482-486 (2015).

Ref 38 Wang, L. et al. Tuning magnetotransport in a compensated semimetal at the atomic 475 scale. Nat. Commun. 6, 8892 (2015).

The author claims different R-T behavior for bilayer and trilayer MoTe₂ suggest band gap opening. But references 37, 38 citations are not objective. For the band gap opening behavior mentioned in ref 37, these flakes are unprotected. The following work proves that all the MoTe₂ samples shows metallic behavior even the thickness is down to 7nm (PHYSICAL REVIEW B 97, 041410(R) (2018)). In addition, there is another work that directly proves that the monolayer and bilayer MoTe₂ samples are metallic and exhibit superconducting transition at low temperature (Nano Lett. 2021, 21, 6, 2505–2511).

Ref 38 Wang, L. et al. Tuning magnetotransport in a compensated semimetal at the atomic 475 scale. Nat. Commun. 6, 8892 (2015). There should be other works PHYSICAL REVIEW B 95, 041410(R) (2017), Nature Physics volume 13, pages677–682 (2017) gives contradictory results for trilayer WTe₂ R-T behavior.

Although the author encapsulated with BN and the fabrication process was carried under inert atmosphere. But the result from Nano Lett. 2021, 21, 6, 2505–2511 are more convincing, because flakes were exfoliated using high-quality crystal and encapsulated in double-layer BN. Also via-contact embedded to avoid the influence of strain.

I am not sure whether it is because of the roughness of the substrate (author spin coated sodium cholate solution and AHM) that introduced strain, or if defects, vacancy or other factors induce different R-T behaviors, if so, then the intrinsic Berry curvature interpretation is controversial.

According to the above reference, I do not agree that the semiconductor behavior is the intrinsic behavior of bilayer MoTe₂, then the subsequent nonlinear Hall and other phenomena will be controversial.

Fig 4d, author did not explain the reason why Linear dependence of $E_{2\omega}$ on $E_{||}$ flip sign. In addition, I found that the figure plotting in this manuscript is too similar with Nature Materials volume 18, pages324–328 (2019), which is a bit embarrassing. It also greatly reduces the innovation of this manuscript.

Reviewer #3 (Remarks to the Author):

The manuscript aims to report on the controlled CVD growth of MoTe₂ bi-layers exhibiting strong non-linear hall effect. On this scope the Authors provide evidence of the MoTe₂ crystalline quality and structure with characterization from different techniques. MoTe₂ electrical properties are measured via a 4 terminal fabricated device on top of micrometer size MoTe₂ domains, from which the hall effect is extracted. DFT ab-initio methods are used to uncover the effects on the Berry curvature related to precise layer number.

Overall, the results appear original and of significance in the field, in particular in relation to the precise control of the layer uniformity and regarding the NHE voltage at fixed input current. However, there are several points to be further addressed in detail requiring major revision before further consideration.

Regarding the CVD growth, the methods needs additional details in terms of the description of the "two-step" process. Missing data are related to the pressure (is it and AP-CVD?), precise reading of the temperature at the positions where the precursors are placed and effective time duration for the single steps; the temperature profile in fig 1a, right seems to be a bit generic and not aligned with the text in the methods section. Sentences such as "MoO₃ powder (10 mg) and solid tellurium powder (50 mg) were placed 5 cm and 15 cm upstream from the substrate, respectively." and "After some time, H₂" assumes that the temperature profile at each position follows an undefined temperature profile, to be specified.

Another point is related to the claim of large-scale growth. Why it is clear that the

Authors could reach the coverage of almost the entire substrate size, obtaining a polycrystalline 1T' MoTe₂ film, most of their characterization and, more importantly, the electrical measurements are considering micrometer size (elongated) mono? domains. Thus, the results shown and the related discussion seems to be more focused on the study of the properties of such micro-sized domains (where orientations are visibly recognized by optical microscopy) more than on the cm scale. I think that this point needs to be clearly stated by the Authors, starting from a revision of the title.

Regarding the physical characterizations, the Authors are strongly encouraged to improve the results shown in the supplemental material. XRD pattern must include at least Miller indexing and a reference pattern; Raman spectra must include phonon modes assignments and a short discussion on the reason of phonon modes shifting in position depending on the number of the layers.

In relation to the DFT study, I suggest to reconsider the period at lines 176-182 in relation to the different panels in fig.3 and the values reported in such graphs; I found a bit difficult to follows the sequence of link to the different panels.

In relation to Fig3e, few lines below I suggest to use "minimu" instead of "peak" for the position of Dxz.

When reporting the electrical measurements, please re-check symbols at lines 225-226, for an obvious inconsistency on the geometric dimensions.

Figures:

- Fig.3 caption "Dotted lines in lower part of (e) and (f) denotes Fermi energy" -> "Dotted lines denote Fermi energy", according to the line traced in each panel.**
- Fig.4b, right vertical axis (red), symbols should be "rho-parallel" component.**

Response to Reviewers' Comments

Reviewer #1 (Remarks to the Author):

In this work, the authors demonstrate CVD synthesis of MoTe₂ with large domain size (up to 500 μm) and precise control of layer numbers at the ultra-thin limit through a layer-by-layer growth. The synthesized crystals are identified using TEM and Raman spectroscopy. Although the large-area MoTe₂ growth has been achieved using CVD method in literature (Adv. Mater. 28, 9526–9531 (2016), Adv. Mater. 29, 1700704 (2017) and ACS Nano 15, 410–418 (2021)), the precise control of bilayer growth was not presented. The achievement presented in the work provides a platform to explore more interesting properties and applications of the material. Particularly, the nonlinear Hall (NLH) effect observed on bilayer and trilayer T_d phase MoTe₂ is interesting and shows the importance of precise thickness control of the growth. More in-depth understanding of some data is required. Below are some questions and comments to be addressed:

Response: We thank the reviewer for recognizing the novelty of our work in growing bilayer MoTe₂ as well as characterizing the NLE responses. In accordance with the reviewer's suggestions, we have performed additional experiments, provided additional experimental details and added more discussions, which we hope will improve the quality of the manuscript for publication.

1. In the paper, the authors described the synthesized flakes as single crystals with domain size of up to 500 μm (line 100). The optical microscope images show smooth and uniform surface of the flakes. However, the resolution of optical microscope is not high enough to identify the grain boundaries (if they exist) in the sample. More structural evidence (e.g., SAED on different location of the same flake) is required to draw such conclusions.

Response: We thank the reviewer for the suggestions. We agree that the grain boundaries cannot be seen in optical microscope images. Therefore, we have performed aberration-corrected scanning transmission electron microscopy (STEM) measurements on the MoTe₂ flakes. First, the as-grown MoTe₂ flakes were transferred onto the TEM grids by dry transfer, which avoid degradation and damage. Then we combed the area and characterized six regions over a length span of several tens of micrometers wide in space by STEM at a single MoTe₂ flake. As shown in

Fig. R1, we could not find any grain boundaries and the atomic arrangements at different regions show the same orientation, and the corresponding fast Fourier transforms (FFT) show the same pattern, both of which confirm the single-crystal nature of the rectangular MoTe₂ flake.

Fig. R1 STEM characterization of bilayer 1T' MoTe₂ flake with different locations. **a**, Optical image of one rectangular bilayer MoTe₂ flake. The scale bar is 5 μm . **b-m**, Atomic STEM images (**c,e,f,i,j,m**) and corresponding FFT patterns (**b,d,g,h,k,l**) at different locations (1-6), respectively, showing the same orientation across the whole flake. All the scale bars are 1 nm.

2. From the optical image, the monolayer growth is preferred along certain directions. What is the reason? What is the stacking order for bilayer and trilayer? Can low frequency Raman signals be observed from the bilayer and trilayer?

Response: The optical image of the monolayer MoTe₂ flakes shows that they are rectangular in shape. Based on the low-magnification STEM image and atomic structures in Fig. R2, we can see that the rectangular reciprocal lattice is characteristic of the 1T' phase. The quasi-1D molybdenum-tellurium zigzag chains are along the *a*-axis of the unit cell and connected by Te atoms in between,

as indicated in the atomic structural model. As the CVD growth process is conducted near the equilibrium state, the zigzag chain direction is the most energy favorable edge (see for example *npj 2D Mater. Appl.* 1, 8, 2017). The monoclinic crystal structure causes the morphology of the 1T' MoTe₂ to be rectangular, and this has also been commonly observed in literatures (*Nano Lett.* 16, 4297–4304, 2016; *Nat. Nanotech.* 12, 1064-1070, 2017 and *ACS Nano* 15, 4213-4225, 2021). Also, the high-resolution atomic structure of the bilayer and trilayer 1T' MoTe₂ show that the stacking order is 1T' stacking, which is characterized by a distorted unit cell. The 1T' phase's symmetry elements include a 2-fold screw axis along the x direction, a horizontal mirror plane, and a resulting inversion center, and these are applicable to all odd-layer 1T' MoTe₂, including the monolayer and trilayer. In contrast, the even-layer (bilayer) does not possess a screw axis and thereby loses an inversion center, leading to the point group C_s (m) with only a mirror plane. Our STEM-HAADF image shows excellent agreement with the bilayer and trilayer simulated model and previous reports (*Nano Lett.* 16, 4297–4304, 2016 and *Nat. Nanotech.* 12, 1064-1070, 2017).

Fig. R2 Structural characterization of bilayer 1T' MoTe₂ single crystals. **a,b**, Top (a) and side (b) views of the crystal structure of 1T' MoTe₂. **c**, Optical image of bilayer 1T' MoTe₂ single crystals. The scale bar is 50 μm. **d**, Low-magnification TEM image of a bilayer 1T' MoTe₂ single crystal covering the TEM grids. The scale bar is 200 nm. **e**, High resolution STEM image showing the atomic structure with being 1T' phase. The scale bar is 1 nm.

Indeed, Raman spectroscopy is a powerful tool to investigate the crystal symmetry, interlayer coupling and layer stacking in 2D materials. For 1T' MoTe₂, it has several characteristic Raman peaks: a prominent peak of A_g mode at $\approx 85\text{ cm}^{-1}$, two A_g modes at $\approx 104\text{ cm}^{-1}$ and $\approx 116\text{ cm}^{-1}$, another prominent A_g mode at $\approx 162\text{ cm}^{-1}$, a B_g mode at $\approx 190\text{ cm}^{-1}$, and two A_g modes at $\approx 258\text{ cm}^{-1}$ and $\approx 269\text{ cm}^{-1}$ (Fig. R3). These Raman features are consistent with theoretical predictions and Raman spectra of exfoliated 1T' MoTe₂ (Nano Lett. 16, 4297–4304, 2016). For the Raman mode at low frequency, the A_g peak at $\approx 13\text{ cm}^{-1}$ would appear for bilayer and trilayer samples (ACS Nano, 15, 2962–2970, 2021). The monolayer MoTe₂ could be stacked in two forms: one is 1T' stacking and the other is T_d stacking. The 1T' and T_d MoTe₂ can be differentiated in interlayer vibration modes, in which the A_g mode at $\approx 116\text{ cm}^{-1}$ and the B_g mode at $\approx 190\text{ cm}^{-1}$ in 1T' MoTe₂ are split into two peaks in T_d phase. Based on this, our CVD-grown MoTe₂ shows clearly the characteristics of 1T' phase, which is also in agreement with the STEM observations. For the layer numbers, the higher energy mode at $\approx 271\text{ cm}^{-1}$ red shifts when the thickness of 1T' MoTe₂ increase from monolayer to bilayer (268 cm^{-1}), and trilayer (265 cm^{-1}). This is because the out-of-plane A_g mode at this energy mode is strongly affected by the interlayer interactions. It is considered that such a frequency drop with increased thickness is possibly due to the enhancement of dielectric screening of the long-range Coulomb interaction for an excitation going from monolayer to few-layer MoTe₂ (J. Am. Chem. Soc. 139, 8396–8399, 2017). Therefore, a fingerprint Raman peak at 269 cm^{-1} distinguishes monolayer MoTe₂ and the peak position redshifts to 267 , and 265 cm^{-1} for bilayer and trilayer, respectively. As shown in Fig. R3, the Raman mapping of this peak shows a very uniform signal at 269 cm^{-1} for monolayer MoTe₂ film (top left), 267 cm^{-1} for bilayer crystals with different orientations (bottom left) and film (middle panel), and 265 cm^{-1} for trilayer films (right panel), suggesting the uniform feature for the MoTe₂ with different layers.

Fig. R3 Optical images and Raman characterization of MoTe₂ with different layers. a,b, Optical image (a) and Raman mappings (b) of MoTe₂ monolayer (top left), second layer with different orientation on top of the monolayer (bottom left), bilayer film (middle), and trilayer films (right). All the scale bars are 2 μm . **c,** Typical Raman spectra of MoTe₂ with different layers, showing the 1T' phase characteristics.

3. For the hall bar devices, the authors claimed that the electrodes are patterned along the a- or b-axis of the crystal (line 197). How is this identified?

Response: As shown in Fig. R2, the lattice is oriented such that the *a*-direction points along the long-axis of the rectangular flake, which is consistent in every flake. Therefore, we can make use of this anisotropic growth to pattern the electrodes along the a-axis of the crystal.

4. Non-linear Hall effect (NLHE) is observed on MoTe₂ flakes in this work, and opposite NLHE magnitude versus conductivity relationships have been observed for bilayer and trilayer MoTe₂ devices. This is attributed to the low electrical conductivity of bilayer MoTe₂ (line 253). However, the Hall measurements are carried out at low temperature region (<100 K), which coincides with the temperature range of “gap opening” in bilayer MoTe₂ (line 199-204, Fig. 4a). In this scenario, it is difficult to attribute the unique NLHE magnitude vs. conductivity relationship solely to the

low magnitude of electrical conductivity. More control experiments are recommended to identify the origin of this relationship. What if the Hall measurement of bilayer MoTe₂ is carried out at higher temperature where the “gap opening” does not occur (i.e., >100 K)? Will the NLHE magnitude show the same trend as current experiment? Considering the electrical resistivity maintains a low value and does not change much from 100 K to 250 K, the higher temperature region could help identify the role of “gap opening” in this process.

Response: We thank the review for the comment. According to the reviewer’s suggestion, we performed more experiments and measured the 2nd order Hall signal of the bilayer MoTe₂ at high temperatures. As shown in Fig. R4 below, we fit the ratio of the $E_{\perp}^{2\omega}$ on E_{\parallel}^2 to get NLH magnitude for bilayer. At 120 K, the NLH magnitude has decreased by more than 50% compared to the value at 100 K, this is consistent with the commonly reported temperature dependence of NLH magnitude (e.g., Nat. Mater. 18(4), 324-328, 2019 and Nat. Nanotech. 16(8), 869-873, 2021). Moreover, the signal noise at 120 K has increased compared to lower temperatures. Therefore, to get more reliable data, we only consider data below 100 K. Similarly, all previous reports (e.g., Nat. Mater. 18(4), 324-328, 2019 and Nat. Nanotech. 16(8), 869-873, 2021) also reported mainly data below 100 K range. After checking reported literatures on MoTe₂, we have changed the description of resistance-temperature relation into ‘increasing resistance with temperature which could be attributed to spin-orbit coupling, gap-opening or Anderson localization’.

Fig. R4 $E_{\perp}^{2\omega}$ of trilayer MoTe₂ as function of square of electric field at different temperatures.

Reviewer #2 (Remarks to the Author):

The authors have studied the nonlinear hall effect NLH in the bilayer and trilayer MoTe₂. By performing transport, PFM measurements they have found interesting thickness dependence ferroelectric domain and second harmonic voltage. They also performed theoretical calculation to claim the major contribution for the large NLH is Berry curvature dipole. However, the experimental results are contradictory to other more convincing works, which is not listed in the reference in this manuscript and this work is not innovative enough, so I do not recommend publishing in Nature Communications. This statement “In Fig. 4a, the increasing resistivity with decreasing temperature suggests a “gap opening” of bilayer sample at low temperature. Previous reports attributed the insulating behavior in few-layer MoTe₂ to either spin-orbit coupling³⁷ or Anderson localization ^{38-40”} Ref 37 Keum, D. H. et al. Bandgap opening in few-layered monoclinic MoTe₂. Nat. Phys. 11, 473 482-486 (2015). Ref 38 Wang, L. et al. Tuning magnetotransport in a compensated semimetal at the atomic 475 scale. Nat. Commun. 6, 8892 (2015). The author claims different R-T behavior for bilayer and trilayer MoTe₂ suggest band gap opening. But references 37, 38 citations are not objective. For the band gap opening behavior mentioned in ref 37, these flakes are unprotected. The following work proves that all the MoTe₂ samples shows metallic behavior even the thickness is down to 7nm (PHYSICAL REVIEW B 97, 041410(R) (2018)). In addition, there is another work that directly proves that the monolayer and bilayer MoTe₂ samples are metallic and exhibit superconducting transition at low temperature (Nano Lett. 2021, 21, 6, 2505–2511). Ref 38 Wang, L. et al. Tuning magnetotransport in a compensated semimetal at the atomic scale. Nat. Commun. 6, 8892 (2015). There should be other works PHYSICAL REVIEW B 95, 041410(R) (2017), Nature Physics volume 13, pages677–682 (2017) gives contradictory results for trilayer WTe₂ R-T behavior. Although the author encapsulated with BN and the fabrication process was carried under inert atmosphere. But the result from Nano Lett. 2021, 21, 6, 2505–2511 are more convincing, because flakes were exfoliated using high-quality crystal and encapsulated in double-layer BN. Also via-contact embedded to avoid the influence of strain. I am not sure whether it is because of the roughness of the substrate (author spin coated sodium cholate solution and AHM) that introduced strain, or if defects, vacancy or other factors induce different R-T behaviors, if so, then the intrinsic Berry curvature interpretation is controversial. According to the above reference, I do not agree that the semiconductor behavior is the intrinsic behavior of bilayer MoTe₂, then the subsequent nonlinear

Hall and other phenomena will be controversial. Fig 4d, author did not explain the reason why Linear dependence of $E_{2\omega}$ on $E_{||}$ flip sign. In addition, I found that the figure plotting in this manuscript is too similar with Nature Materials volume 18, pages 324–328 (2019), which is a bit embarrassing. It also greatly reduces the innovation of this manuscript.

Response: We thank the reviewer for his critical comments. The gap opening is only an alternative interpretation of the electrical results, among several others. We would like to remind that **the main focus and novelty of our paper is on the deterministic growth of bilayer MoTe₂ single crystals, as well as the characterization of its ferroelectric and NLE properties compared to thicker layer.** We do not think it is right to claim that our paper is not novel simply on the basis one interpretation of a data presented, and on the way the data is plotted (standard way of plotting NLE results in literatures). Nobody has reported the deterministic growth of bilayer MoTe₂, let alone characterized the ferroelectric and NLE properties.

In the article we ascribe the increasing resistance of bilayer MoTe₂ with decreasing temperature to ‘gap-opening’. It is worth noting that the origin is far from clear and even controversial. Nonetheless, in view of the referee’s negative comment on this, we will revise this statement to adopt a more balanced view that there could be several origins to this.

In the revised paper, we have revised the statement to this *‘the upswing in resistance with temperature for few-layer samples has been reported previously (Ref. 1, Nat. Phys. 11, 482-486, 2015; Ref. 2: Nat. Comm. 6, 8892, 2015; Ref. 3: 2D Mater. 5, 031010, 2018; Ref. 4: Nat. Comm. 10, 2044, 2019), several origins such as spin-orbit coupling induced gap (Ref. 1) and Anderson localizations (Ref. 2 and 3) have been suggested’.*

In Ref. 1, few-layer MoTe₂ is shown to have a bandgap ascribed to strong spin-orbit coupling.

In Ref. 2, the transport behavior of WTe₂ goes through metal-insulator transition when the vertical thickness of the sample is reduced. The authors found that the mobility of the charge carriers are 2-3 orders of magnitude smaller than that of thicker layers and thus disorder should be the cause. They argue that the carriers are Anderson localized. The metal-insulator transition in few-layer MoTe₂ is also explained as enhanced charge carrier localization in Ref. 3.

Additionally, we need to explain that the paper cited by reviewer (Phys. Rev. B 97, 041410(R), 2018) **studied MoTe₂ that are much thicker than ours, and they reported that these thicker**

crystals show metallic behavior down to 7 nm. Here, our samples are much thinner: the bilayer is 1.4 nm and the trilayer is 2.1 nm. Specifically, our trilayer MoTe₂ indeed shows metallic transport behavior. For this thickness regime, it is very difficult to exfoliate mechanically or grow high quality crystals, thus on this basis alone, our growth and characterization part is highly novel. According to the cited papers (Phys. Rev. B 95, 041410(R), 2017 and Nat. Phys. 13, 677–682, 2017), **the trilayer devices all show metallic behavior, consistent with our trilayer’s R-T behavior. The main difference occurs for the bilayer sample that shows the resistance upswing at lower temperature. In other words, it is not correct to say that we claim gap-opening for all, for thickness that is trilayer and thicker, we observed metallic characteristics, consistent with what the referee expects.**

We have also checked the literature (Nano Lett. 21, 6, 2505–2511, 2021) as mentioned by reviewer. The main difference between our work and the *Nano Lett.* Paper is that the authors used hBN to fully encapsulate the MoTe₂ flake. In our case, our sample was only covered by hBN on top to provide protection from degradation and the SiO₂/Si substrate has an influence on MoTe₂. In surveying literature, we found that the substrate influences the device behavior of the MoTe₂ sample significantly, in which few-layer MoTe₂ has been reported to exhibit insulating or ‘gapped/band-splitting’ behaviors depending on the substrate. For instance, in the Fig. 1d of *Nature Comm. 10, 2044, 2019, the R-T relation of the 2 nm-thick MoTe₂ (can be considered as bilayer) also shows an upswing below 50 K before transition into superconducting state.* We observed that the authors in this reference used SiO₂/Si substrate similar to ours. Additionally, monolayer MoTe₂ exhibit semi-metallic behavior with large band overlap when grown on bilayer graphene (APL Mater. 6, 026601, 2018) but a weak overlap with a potential gap-opening when exfoliated on gold substrates (Phys. Rev. Mater. 2, 104004, 2018). Therefore, we can infer that the electronic band structures of few-layer MoTe₂ is very sensitive to the substrate.

Based on the above analysis, our bilayer sample may have more skew scattering from SiO₂/Si substrate compared to hBN. This point is also reflected in our discussion of the NLH magnitude in the paper in which we stated that the NLE has multiple contributions (i.e., skew scattering etc), and not limited to intrinsic Berry curvature. Due to low conductivity of the bilayer, the relation of NLH magnitude with conductivity is complex and cannot be quantified analytically at the present

stage. In contrast, for metallic trilayer MoTe₂, the linear relation in Fig. 5d clearly agrees well with both theory and published literatures.

For Fig. 4d, the sign switching of the linear dependence is to support the 2nd order origin of the Hall signal at each temperature points, which is described in main text: ‘The second order Hall signal switches sign when the current orientation and the Hall probes connections are reversed simultaneously. This sign reversal behavior further consolidates its second order origin because linear Hall effect would vanish under Onsager reciprocal relation while second order one does not need to vanish.’

The way we plot and display the NLE data is in line with standard practice, and this way of reporting data is also shared by authors of related work in *Nature Nanotech.* 16, 869-873, 2021, *Nature Mater.* 18, 324-328, 2019, *Nature Phys.* 2022, DOI: 10.1038/s41567-022-01606-y, in its extended figures, *Nature Nanotech.* 17, 378-383, 2022, and *Nature Comm.* 12, 698, 2021. All these works have similar analysis and convention in plotting to make it convenient for readers to follow.

Reviewer #3 (Remarks to the Author):

The manuscript aims to report on the controlled CVD growth of MoTe₂ bilayers exhibiting strong non-linear Hall effect. On this scope the Authors provide evidence of the MoTe₂ crystalline quality and structure with characterization from different techniques. MoTe₂ electrical properties are measured via a 4 terminal fabricated device on top of micrometer size MoTe₂ domains, from which the Hall effect is extracted. DFT ab-initio methods are used to uncover the effects on the Berry curvature related to precise layer number. Overall, the results appear original and of significance in the field, in particular in relation to the precise control of the layer uniformity and regarding the NHE voltage at fixed input current. However, there are several points to be further addressed in detail requiring major revision before further consideration.

Response: We thank the reviewer for highlighting the originality and significance of our work. According to the reviewer’s following requirements, we have provided more experimental details and performed more studies to improve the quality of the manuscript.

Regarding the CVD growth, the methods need additional details in terms of the description of the "two-step" process. Missing data are related to the pressure (is it and AP-CVD?), precise reading of the temperature at the positions where the precursors are placed and effective time duration for the single steps; the temperature profile in fig 1a, right seems to be a bit generic and not aligned with the text in the methods section. Sentences such as "MoO₃ powder (10 mg) and solid tellurium powder (50 mg) were placed 5 cm and 15 cm upstream from the substrate, respectively." and "After some time, H₂" assumes that the temperature profile at each position follows an undefined temperature profile, to be specified.

Response: We thank the reviewer for pointing out the missing information. The whole CVD reaction is operated at ambient pressure. The temperature profile in Fig. 1a only indicates the temperature of the SiO₂/Si substrate positions. As the MoO₃ powder and solid tellurium powder were placed 5 cm and 15 cm upstream from the substrate, the temperature at the two locations are 670 and 550 °C, respectively. We have revised the Methods session and make the experimental details more clearly.

Another point is related to the claim of large-scale growth. Why it is clear that the Authors could reach the coverage of almost the entire substrate size, obtaining a poly-crystalline 1T' MoTe₂ film, most of their characterization and, more importantly, the electrical measurements are considering micrometer size (elongated) mono domains? Thus, the results shown and the related discussion seems to be more focused on the study of the properties of such micro-sized domains (where orientations are visibly recognized by optical microscopy) more than on the cm scale. I think that this point needs to be clearly stated by the Authors, starting from a revision of the title.

Response: We thank the reviewer for the comment. Using the two-step CVD methods, large-scale single crystal domains and centimeter-sized polycrystalline film could be fabricated. As grain boundaries could affect the electronic properties, we selected single crystal MoTe₂ domain for the device measurement to observe the intrinsic properties of the CVD-grown MoTe₂. We found that bilayer MoTe₂ produces the largest second harmonic output voltage among the thicker crystals. As we mainly focused on the nonlinear Hall effect on the MoTe₂ single crystals, we have revised the title as follows: "Growth of Bilayer MoTe₂ Single Crystals with Strong Non-Linear Hall Effect".

Future work will aim to grow large area single crystal on sapphire and do large array device measurements, which is currently outside the scope of the present work.

Regarding the physical characterizations, the Authors are strongly encouraged to improve the results shown in the supplemental material. XRD pattern must include at least Miller indexing and a reference pattern; Raman spectra must include phonon modes assignments and a short discussion on the reason of phonon modes shifting in position depending on the number of the layers.

Response: We thank the reviewer for the suggestions. We have included the Miller indexing for each peak in XRD patterns and a reference pattern for comparison. As shown in Fig. R5, there are only four main diffraction peaks corresponding to the (002), (004), (006), (008) planes of the MoTe₂ crystals, which is in good agreement with the reference pattern for monoclinic 1T' phase. Obviously, all prominent diffraction peaks are indexed to the {001} family planes, suggesting that the *c*-axis of the as-grown film is perpendicular to the growth substrate and that the growth is highly textured.

Fig. R5. Powder XRD patterns of the large-scale 1T' MoTe₂ films. The blue curve shows the experimental data for 1T' MoTe₂ films while the red curve shows the calculated XRD patterns for reference.

Also, we have included the phonon modes in all the peaks of the Raman spectrum. 1T' MoTe₂ crystals have several characteristic Raman peaks: a prominent peak of A_g mode at $\approx 85 \text{ cm}^{-1}$, two A_g modes at $\approx 104 \text{ cm}^{-1}$ and $\approx 116 \text{ cm}^{-1}$, another prominent A_g mode at $\approx 162 \text{ cm}^{-1}$, a B_g mode at $\approx 190 \text{ cm}^{-1}$, and two A_g modes at $\approx 258 \text{ cm}^{-1}$ and $\approx 269 \text{ cm}^{-1}$. These Raman features are consistent

with theoretical predictions and Raman spectra of exfoliated 1T' MoTe₂. The higher energy mode at $\approx 271 \text{ cm}^{-1}$ red shifts when the thickness of 1T' MoTe₂ increase from monolayer to bilayer (268 cm^{-1}), and trilayer (265 cm^{-1}). This is because the out-of-plane A_g mode at this energy mode is strongly affected by the interlayer interactions. The red shift of the Raman peak with increased thickness is due to the enhancement of dielectric screening of the long-range Coulomb interaction going from monolayer to few-layer MoTe₂ (J. Am. Chem. Soc. 139, 8396–8399, 2017). Therefore, the fingerprint Raman peak at 269 cm^{-1} can be used to identify monolayer MoTe₂ and the peak position redshifts to 267 , and 265 cm^{-1} for bilayer and trilayer, respectively. As shown in Fig. R6, the Raman mapping of this peak shows a very uniform signal at 269 cm^{-1} for monolayer MoTe₂ film (top left), 267 cm^{-1} for bilayer crystals with different orientations (bottom left) and film (middle panel), and 265 cm^{-1} for trilayer films (right panel), suggesting the uniform feature for the MoTe₂ with different layers.

Fig. R6 Optical images and Raman characterization of MoTe₂ films with different layers. a,b, Optical image (a) and Raman mappings (b) of MoTe₂ monolayer (top left), second layer with different orientation on top of the monolayer (bottom left), bilayer film (middle), and trilayer films (right). All the scale bars are $2 \mu\text{m}$. c, Typical Raman spectra of MoTe₂ with different layers, showing the 1T' phase characteristics.

In relation to the DFT study, I suggest to reconsider the period at lines 176-182 in relation to the different panels in fig.3 and the values reported in such graphs; I found a bit difficult to follows the sequence of link to the different panels.

Response: We thank the reviewer for the comment. We have changed the alphabetical orders in Fig. 3 and revised the related sentences as follows: “For the bilayer, four crossing points between the lowest and the second-lowest conduction bands are found at $(\pm 0.2034 \frac{2\pi}{a}, \pm 0.0143 \frac{2\pi}{b})$ in the Brillouin zone (upper inset of Fig. 3a). The Berry curvature near the crossing points calculated by the fixed-number-of-occupied-bands scheme (See Supplementary Section IV) is shown in Fig. 3d. While for the trilayer, anti-crossing points between the second- and the third-lowest conduction bands are found at $(\pm 0.1149 \frac{2\pi}{a}, 0)$ as highlighted in the inset of Fig. 3b. The Berry curvature near the anti-crossing point of trilayer is shown in Fig. 3f.”

In relation to Fig. 3e, few lines below I suggest to use "minimum" instead of "peak" for the position of D_{xz} . When reporting the electrical measurements, please re-check symbols at lines 225-226, for an obvious inconsistency on the geometric dimensions.

Response: We thank the reviewer for this suggestion. When considering only Fig. 3e, the term "minimum" is not a problem. However, "peak" is a safe choice in this case because the sign of the berry curvature D_{xz} component is not determined in practice. For example, when we rotate the sample by 180 degrees around the z-axis, the sign of D_{xz} is inverted. Usually, experiments do not distinguish the orientation of the sample at this level (e.g., the $\pm a$ directions). Therefore, only the absolute value and whether sign change occurs are meaningful. The "minimum" can be "maximum" in another setting. On the other hand, "peak" can be used for both the maximum and minimum in this case.

We have also changed the symbols as “Based on the Hall bar geometry, we can obtain electric fields for each quantity: $E_{\perp}^{2\omega} = \frac{V_{\perp}^{2\omega}}{L_{\perp}}$ and $E_{\parallel} = \frac{V_{\parallel}}{L_{\parallel}}$ where $L_{\parallel} \approx 2.4 \mu\text{m}$ and $L_{\perp} \approx 1.7 \mu\text{m}$ are longitudinal and transverse lengths of the bilayer Hall bar device.”.

Figures:

- Fig.3 caption "Dotted lines in lower part of (e) and (f) denotes Fermi energy" -> "Dotted lines denote Fermi energy", according to the line traced in each panel.

- Fig.4b, right vertical axis (red), symbols should be "rho-parallel" component.

Response: We thank the reviewer for the careful inspection. We have replaced the Fig. 3 caption by “Dotted lines in g and h denote Fermi energy of 0.28 eV and 0.24 eV for bilayer and trilayer sample, respectively”, and changed the right vertical axis of Fig. 4b to ρ_{\parallel} .

REVIEWER COMMENTS

Reviewer #1 (Remarks to the Author):

The authors have addressed all my comments.

Reviewer #2 (Remarks to the Author):

I am glad that the author can organize the literature and change some controversial conclusions. In fact, comparing Ref. 1, Nat. Phys. 11, 482-486, 2015 with Nano Lett. 21, 6, 2505-2511, 2021, the few-layer samples have different electrical transport behaviors, I suggest the authors delete the citation of Ref 1 and related discussions about it, which was not protected with BN at the time. In fact, for these type of materials (MoTe₂ WTe₂), whether it is covered with BN or not greatly affects the behavior of electrical transport. For example, in the Ref 2 Nat. Comm. 6, 8892, 2015 mentioned by the author, 4L sample is a semiconductor behavior without covering with BN. But for PHYSICAL REVIEW B 95, 041410(R) (2017), 3L is still a metallic behavior. The substrate will also affect its electrical transport behavior(ACS Appl. Mater. Interfaces 2017, 9, 27, 23175-23180. Therefore, the optimized device fabrication process gives more convincing results for few-layer samples, especially for bilayer samples. Nature volume 565, pages337-342 (2019) . The growth of a large-area bilayer sample is a breakthrough in this field. But at this point, I recommend against publication in Nature communications. Before further consideration, the authors should address some concerns that I list in the following:

If the author can fully encapsulate it with BN, the experimental data will be more convincing. If the experimental conditions are limited and the author can only cover BN on the top surface, I think authors should at least add the d.c component of the transverse voltage VDC and compare with second harmonic transverse voltage V_{xy}^{2nd} , also longitudinal response V_{xx}^{2nd} should be compared with V_{xy}^{2nd} data.

I suggest authors combine the above data with symmetry analysis to make the experimental results more convincing.

Reviewer #3 (Remarks to the Author):

In the revised version the Authors satisfactory replied to my previous comments. The Authors made also corrections, modifications and integration in the main text, supplementary info and figures addressing the my previous requests.

In my opinion, at the best of my knowledge on the topic, the manuscript has the degree of novelty and the scientific soundness to be accepted for publication in this journal.

Response to Reviewers' Comments

Reviewer #1 (Remarks to the Author):

The authors have addressed all my comments.:

Response: We thank the reviewer very much for the positive comments.

Reviewer #2 (Remarks to the Author):

I am glad that the author can organize the literature and change some controversial conclusions. In fact, comparing Ref. 1, Nat. Phys. 11, 482-486, 2015 with Nano Lett. 21, 6, 2505–2511, 2021, the few-layer samples have different electrical transport behaviors, I suggest the authors delete the citation of Ref 1 and related discussions about it, which was not protected with BN at the time.

In fact, for these types of materials (MoTe_2 or WTe_2), whether it is covered with BN or not greatly affects the behavior of electrical transport. For example, in the Ref 2 Nat. Comm. 6, 8892, 2015 mentioned by the author, 4L sample is a semiconductor behavior without covering with BN. But for PHYSICAL REVIEW B 95, 041410(R) (2017), 3L is still a metallic behavior. The substrate will also affect its electrical transport behavior (ACS Appl. Mater. Interfaces 2017, 9, 27, 23175–23180. Therefore, the optimized device fabrication process gives more convincing results for few-layer samples, especially for bilayer samples. Nature volume 565, pages337–342 (2019).

The growth of a large-area bilayer sample is a breakthrough in this field. But at this point, I recommend against publication in Nature Communications. Before further consideration, the authors should address some concerns that I list in the following:

If the author can fully encapsulate it with BN, the experimental data will be more convincing.

If the experimental conditions are limited and the author can only cover BN on the top surface, I think authors should at least add the d.c component of the transverse voltage V_{DC} and compare with second harmonic transverse voltage V_{xy}^{2nd} , also longitudinal response V_{xx}^{2nd} should be compared with V_{xy}^{2nd} data.

I suggest authors combine the above data with symmetry analysis to make the experimental results more convincing.

Response: We thank the reviewer for his suggestions. In accordance with the reviewer's suggestions, we have performed additional experiments and added more discussions to improve the quality of the manuscript for publication.

First, we agree with the reviewer that the presence or absence of hBN encapsulation will have effects on the behavior of the electrical transport. According to the reviewer's suggestion, we have deleted Ref. 1 and related discussions as the sample in that literature was not protected with hBN.

We also agree with the reviewer that the substrate could affect the electrical transport behavior. Fully encapsulating the MoTe₂ with hBN could reveal the intrinsic electronic properties of MoTe₂. However, in our work, the MoTe₂ is grown on SiO₂/Si substrate directly at high temperature, and the binding force is sufficiently strong that we could not delaminate the MoTe₂ flake without causing tear to appear. It is worth noting that the highlights of our work are the *fabrication of large-area bilayer MoTe₂ films and observation of its large second harmonic output voltage on the grown SiO₂/Si substrate*. Although the SiO₂/Si substrate may affect the electronic properties of the bilayer MoTe₂, *we still observed the robust second harmonic properties on the SiO₂/Si substrate, which is very useful for practical applications*. Moreover, SiO₂/Si is a common and necessary substrate for semiconductor industry. Obtaining the high performance of MoTe₂ directly on SiO₂/Si substrate is critical for pushing into the industry application.

Then, as the bilayer MoTe₂ is unstable in air, to demonstrate the electronic properties on its grown SiO₂/Si substrate, top encapsulation with hBN is required to avoid any degradation of the bilayer MoTe₂ sample. We fabricated more devices and provided additional data to further illustrate the 2nd order origin of the Hall signal. Similar to Fig. 4d in main text, the $E_{xx}^{2\omega}$ (2nd order longitudinal electric field strength) and $E_{xy}^{2\omega}$ (2nd order transverse (Hall) electric field strength) is plotted against E_{xx} (1st order longitudinal electric field) in Fig. R1. It is clearly shown that the 2nd order Hall signal dominates over the longitudinal counterpart, which also rules out the possibility of thermal-induced 2nd order effect which is isotropic.

Fig. R1. Second order Hall (blue) and longitudinal (red) electric field as function of square of longitudinal electric field in a bilayer device.

Also, we observed that the phase of 2nd Hall voltage with respect to longitudinal voltage has a 90-degree phase shift, as shown in Fig. R2. The consistent phase further supports the 2nd order origin since a straightforward relation of 2nd harmonic signal should follow as:

$$V^{\text{second-order}} \propto [I_0 \sin(\omega t)]^2 = I_0^2 [1 + \sin(2\omega t - \pi/2)]/2$$

Fig. R2. Phase of 2nd order Hall signal as function of longitudinal voltage.

For the symmetry analysis, in the main text, we focused on the 2nd harmonic Hall signal where the current is applied along a -axis and we detected the voltage difference along b -axis (as denoted in the inset a in Fig. R3). This measurement setup is inspired by the theory that the Berry curvature dipole lies along a -axis due to mirror symmetry in MoTe₂. Here, we also measured the 2nd harmonic Hall signal in the opposite geometry. In this opposite geometry, the current passes along b -axis and we detected the voltage along a -axis (as denoted in the inset b in Fig. R3). When plotted together, we can find the 2nd harmonic response acquired in opposite geometry is much weaker. This further confirms the symmetry selection rule.

Fig. R3. Second order Hall electric field as function of applied current for normal geometry (black) and opposite geometry (blue), respectively.

Finally, we have tried to measure the DC component of the Hall signal. Nevertheless, considering the μV range of the DC signal which accompanies with an equivalent-size AC part, it is challenging to get highly resolved DC voltages without compromising the fidelity of the signal. Actually by careful scrutinization of the related literature and to the best of our knowledge, we find that there are no reported DC voltages in all the literatures expect for one *Nature Materials* paper (Nat Mater, 18, 324-328, 2019) paper regarding the in-plane nonlinear Hall effects. For instance, one *Nature* paper (Nature, 565, 337–342, 2019) clearly demonstrated that they only focused on the second-harmonic signals because that allows them to use the lock-in technique, which greatly enhances the measurement sensitivity and precision. In other words, to measure the DC signal, we have to change from the lock-in technique to DC voltmeter, which will suffer from noise and sacrifice precision.

Meanwhile, for the mentioned *Nature Materials* paper, the DC signal was only reported in its Fig. 2c with a four-layer device. Specifically, we notice that the slope of linear relation in its Fig. 2c is 6 orders of magnitude larger than that in its Fig. 2b (five-layer device). Considering that $E_{xy}^{2\omega}$ and

E_{xx} are related to $V_{xy}^{2\omega}$ and V_{xx} just by a geometric factor, the resulted $E_{xy}^{2\omega}/E_{xx}^2$ would also be around 6 orders of magnitude larger than the result in its Fig. 2b and Fig. 4a. Given that they did not show DC measurement for five-layer device (which is the main device they acquired data except for Fig. 2c and it has similar $E_{xy}^{2\omega}/E_{xx}^2$ magnitude as ours), it is reasonable to argue that only devices with extraordinarily large $E_{xy}^{2\omega}/E_{xx}^2$ magnitude can manifest detectable DC voltages with reliable precision. In our work, the $E_{xy}^{2\omega}/E_{xx}^2$ value is much smaller than that in this Nat Mater Paper, and thus the DC voltage is noisy, which is similar with the mentioned Nature paper (Nature, 565, 337–342, 2019).

Finally, we would like to thank the reviewer for raising these questions and help understand and present our data in a better way.

Reviewer #3 (Remarks to the Author):

In the revised version the Authors satisfactory replied to my previous comments. The Authors made also corrections, modifications and integration in the main text, supplementary info and figures addressing my previous requests.

In my opinion, at the best of my knowledge on the topic, the manuscript has the degree of novelty and the scientific soundness to be accepted for publication in this journal.

Response: We thank the reviewer very much for recognizing the novelty of our work.

REVIEWERS' COMMENTS

Reviewer #2 (Remarks to the Author):

The authors have addressed my comments.